# A Matter of Margins in Oral Cancer—How Close Is Enough?

**DOI:** 10.3390/cancers16081488

**Published:** 2024-04-12

**Authors:** Mateusz Szewczyk, Jakub Pazdrowski, Piotr Pieńkowski, Bartosz Wojtera, Barbara Więckowska, Paweł Golusiński, Wojciech Golusiński

**Affiliations:** 1Department of Head and Neck Surgery, Poznań University of Medical Sciences, 61-701 Poznań, Poland; jakub.pazdrowski@wco.pl (J.P.); piotr.pienkowski@wco.pl (P.P.); bartosz.wojtera@wco.pl (B.W.); wojciech.golusinski@wco.pl (W.G.); 2The Greater Poland Cancer Center, 61-866 Poznań, Poland; 3Department of Computer Science and Statistics, Poznań University of Medical Sciences, 61-701 Poznań, Poland; barbara.wieckowska@ump.edu.pl; 4Department of Otolaryngology and Maxillofacial Surgery, University of Zielona Góra, 65-417 Zielona Góra, Poland; pgolusinski@uz.zgora.pl

**Keywords:** oral, oral cancer, margins, close margins

## Abstract

**Simple Summary:**

In patients with oral cancer, the risk factors for local, regional, and distant recurrence according to margin status have not been well established. The aims of the present study were to identify a margin cut-off point for improved survival in patients with close margins. A retrospective review of adult patients treated surgically at our centre for primary oral cavity squamous cell cancer from 2009 to 2021 was carried out. Margins (mucosal and deep) were classified as positive (<1 mm), close (1 to 4.9 mm), or clear (>5 mm). A total of 326 patients (210 men; 64.4%) were included. Margin status was as follows: close (n = 168, 51.5%), clear (n = 83, 25.4%), and positive (n = 75, 23.0%). The optimal cut-off for disease-free survival was a deep margin > 3 mm. In the close margin group, survival was significantly better in patients with a deep margin > 3 mm. Moreover, in this subset of patients, survival was comparable to the outcomes observed in patients with clear margins, suggesting that surgical margins smaller than the standard cut-off (5 mm) may be sufficient in certain well-defined cases.

**Abstract:**

In patients with oral cancer, the risk factors for local, regional, and distant recurrence according to margin status have not been well established. We aimed to determine the risk factors for recurrence by margin status and to identify a margin cut-off point for improved survival in patients with close margins. We retrospectively reviewed adult patients treated at our centre from 2009 to 2021 for primary oral cancer. Margins were classified as positive (<1 mm), close (1 to 4.9 mm), or clear (>5 mm). Univariate and multivariate analyses were performed. A total of 326 patients (210 men) were included. The mean age was 59.1 years. Margin status was close (n = 168, 51.5%), clear (n = 83, 25.4%), or positive (n = 75, 23.0%). In the univariate analysis, positive surgical margins (HR = 7.53) had the greatest impact on distant failure. Positive surgical margins—without nodal involvement—had the greatest impact on the risk of distant failure. In the close margin group, the optimal cut-off for disease-free survival (AUC = 0.58) and overall survival (AUC = 0.63) was a deep margin > 3 mm, with survival outcomes that were comparable to the clear margin group. These finding suggest that margins < 5 mm may be sufficient in certain well-defined cases. Prospective studies are warranted to confirm these findings.

## 1. Introduction

Oral cancer is the most common type of head and neck cancer [1,2]. In most cases, the treatment of choice is surgical resection of the primary tumour with negative resection margins [3,4]. Positive surgical margins are one of the main risk factors for recurrence following surgery, which is why complete resection with a sufficient surgical margin is essential to avoid treatment failure [5,6]. While it is clear that a sufficient surgical margin is essential to ensure local control, the definition of clear margins has long been controversial. In oral cancer, the most widely accepted clearance margin is 5 mm, as supported by a study carried out by the Royal College of Pathologists (RCP) [7] and a survey of head and neck surgeons conducted by the American Head and Neck Society (AHNS) [8]. The definition of a “close” margin is also challenging, but most scientific societies—including the RCP, the AHNS, and the European Head and Neck Society (EHNS) [9]—define this as a margin from 1 to 5 mm.

Margin status is important because it determines the indication for adjuvant treatment. Moreover, several studies have suggested that smaller margins (i.e., ≤5 mm) may not significantly influence overall survival or local recurrence [10,11,12,13,14]. One recent study found that clinical outcomes in patients with surgical margins > 1 mm were comparable to those observed in patients with margins > 5 mm [15]. Although adjuvant treatment (radiotherapy or chemoradiotherapy) is generally indicated in patients with close margins, good oncological outcomes can be achieved in well-selected patients without adjuvant treatment [16]. Patients with close margins represent a highly heterogenous subgroup, with some patients presenting more adverse clinicopathological features than others [12,17,18]. In fact, the data show that some patients with close margins achieve clinical outcomes that are similar to those observed in patients with clear margins, whereas others experience outcomes that are more in line with those found in patients with involved margins [19,20]. While numerous factors play a role in determining treatment outcomes in patients with close margins, there is little doubt that the surgical margin is a key factor [15]. However, as the findings of the studies described above indicate, margins < 5 mm may be sufficient in some cases. In this regard, it would be valuable to identify a cut-off point that would differentiate between patients with close margins in terms of better or worse survival.

In patients with oral cancer, the risk factors for local, regional, and distant recurrence according to margin status have not been well established. In this context, we conducted the present retrospective study to determine the risk factors for recurrence according to margin status and to identify a margin cut-off point for improved survival in patients with close margins.

## 2. Materials and Methods

This was a retrospective review of adult patients (age 18 or older) treated surgically for primary oral cavity squamous cell cancer (OCSCC) from 2009 to 2021 at our institution. Due to the retrospective nature of the study, approval from the ethics committee was not required.

We included all patients who underwent primary surgery for OCSCC ± adjuvant treatment during the study period except for those who met any of the following exclusion criteria: recurrent OCSCC; second primary malignancy; synchronous primary malignancy; follow up < 24 months (unless recurrence or death occurred earlier); and primary treatment other than surgery.

We evaluated the following variables: demographics (age, sex, smoking status, and general health status [Charlson Comorbidity Index]) and pathologic factors (T stage; N stage; disease stage; primary tumour margin status; perineural invasion (PNI); lymphovascular invasion (LVI); extranodal extension (ENE); and adjuvant treatment [radiotherapy or chemoradiotherapy]).

All patients underwent surgical resection (*en bloc* intent) with a macroscopically-free (1 cm) tumour margin resection. Based on the patient’s preoperative tumour stage and diagnostic evaluation, simultaneous neck dissection was carried out when indicated. Frozen section margin evaluation was performed in all cases. In patients with positive margins, the tumour bed was re-resected, but the patient was still treated as if the final margin was positive.

Tumours were staged according to the 7th edition of the American Joint Committee on Cancer (AJCC) [21]. Re-staging could not be performed because most of the histopathologic reports did not provide data on the depth of invasion. All patients were presented to a multidisciplinary tumour board to determine the indication for adjuvant treatment (standard radiotherapy protocol was 60–66 Gy administered daily Monday–Friday for 6 to 7 weeks; the chemotherapy regimen consisted of single-agent cisplatin (100 mg/m^2^) administered every 3 weeks). Recurrences were documented based on clinical, histopathological, and/or radiological examination. Failure was classified as local, regional, or distant. New tumours located ≥ 2 cm from the primary tumour were classified as second primary tumours. Disease-free survival (DFS) was defined as the time elapsed (in months) from the date of surgery until recurrence. Overall survival (OS) was defined as the time from the date of surgery to the last follow up or death.

Data on the surgical margins—including the distance to the closest margin, the mucosal margin, and the deep margin—were extracted from the pathological reports. Margins were classified as positive (<1 mm), close (1 to 4.9 mm), or clear (>5 mm).

### Statistical Analysis

We used the Cox proportional hazards model to assess the association between risk factors and DFS and OS. This analysis was performed twice. First, we determined the crude hazard ratio (HR) with 95% confidence intervals (CI). Then, on a multivariate analysis, we adjusted the HR for the other variables analysed. To avoid redundancy, we excluded from the corrected model all variables that were highly correlated (C-Pearson’s correlation coefficient > 0.5). The Kaplan–Meier curves are shown in the figures. The distribution of risk factors according to surgical margin status (positive, close, clear) was compared using the chi-square test with the Benjamini–Hochberg correction for multiple comparisons. In the close margin group, we created receiver operating characteristic (ROC) curves for DFS (to evaluate recurrence) and OS (to evaluate mortality). The ROC curves were used to assess the possibility of determining the optimal cut-off point according to margin size. The size of the AUC was examined by DeLong’s method. For statistically significant results, we determined the cut-off point based on the Youden index. We then compared the survival curves according to margin status for this “optimal” cut-off point (<3 mm) for four curves: clear, close (≤ or > the cut-off point), and positive margins. For this comparison, we used the log-rank test with HR and 95% CI. A significance level of alpha = 0.05 was used for the analyses. The PQStat (v1.8.6) and R (v4.3.3; survminer package) software packages were used for all calculations.

## 3. Results

### 3.1. Demographic Data

The final analysis included 326 patients, most of whom were males (n = 210, 64%). The mean age of the sample was 59.1 years (range, 23–97). Most patients were active smokers (n = 190; 60%).

The most common primary tumour sites were the tongue (n = 155; 47.5%) followed by the floor of mouth (n = 111; 34.0%). Adjuvant treatment was administered in 263 cases (80%); of these, 178 (54%) received radiotherapy alone and 85 (26%) received concurrent radiochemotherapy.

A total of 147 (45%) patients developed recurrent disease, which was distributed by location as follows: local (n = 53, 16.2%), regional (n = 33, 10.1%), locoregional (n = 17, 5.2%), distant (n = 21, 6.4%), and locoregional/distant (n = 8, 2.4%). Fifteen patients (4.6%) developed a second primary tumour.

The margin status in the 326 patients was as follows: positive (n = 75, 23.0%), close (n = 168, 51.5%), and clear (n = 83, 25.4%). In patients with close margins, the mean mucosal margin was 2.98 mm (standard deviation [SD], 1.67) and the mean deep margin was 3.27 mm (SD: 2.05).

The clinical and demographic data of the sample are shown in Table 1. Figure 1 and Figure 2 show the Kaplan–Meier curves with DFS and OS stratified by margin status.

### 3.2. Risk Factors for Local, Regional, and Distant Recurrence

Table 2 shows the results of the univariate and multivariate analyses of risk factors for recurrence. Importantly, the risk of local recurrence was statistically significant only in patients with positive margins (HR = 2.75) but not in those with close or clear margins. Similarly, positive margins (but not close or clear margins) were a significant risk factor for both regional and distant recurrence (HR = 2.56 and HR = 7.53, respectively).

### 3.3. Local Recurrence

In the univariate analysis, the risk of local recurrence was highest in patients with nodal disease (HR = 3.54), advanced disease (HR = 3.32), positive margins (HR = 2.75), LVI (HR = 2.94), PNI (HR = 2.09), ENE (HR = 1.91), and advanced T stage (hazard ratio [HR] = 1.75). Not smoking was protective for local recurrence (HR = 0.58). In the multivariate analysis, three variables remained statistically significant for local recurrence: nodal disease (HR = 3.39), advanced disease (HR = 3.37), and positive margins (HR = 2.02).

### 3.4. Regional Recurrence

In the univariate analysis, the following variables were associated with a greater risk of regional recurrence: LVI (HR = 3.67), nodal disease (HR = 3.66), advanced disease (HR = 3.6), PNI (HR = 3.58), ENE (HR = 2.82), positive surgical margins (HR = 2.56), and high tumour grade (HR = 2.16). Not smoking was a protective factor (HR = 0.53). In the multivariate analysis, four factors remained significant: nodal disease, advanced disease, PNI, and ENE.

### 3.5. Distant Recurrence

In the univariate analysis, the following variables were associated with a higher risk of distant recurrence: positive surgical margins (HR = 7.53), nodal disease (HR = 5.94), LVI (HR = 4.74), advanced disease (HR = 3.79), and ENE (HR = 3.46). Adjuvant treatment was a protective factor (HR = 0.13). In the multivariate analysis, nodal disease, LVI, and positive surgical margins all remained significant (Table 2).

### 3.6. Risk Factors for Recurrence Based on Margin Status (Positive, Close, or Clear)

Table 3 shows the clinical and tumour-related risk factors according to margin status. As the table shows, the percentage of patients with locally advanced disease was significantly higher in the positive margin group (33.3%) than in the close (19.6%; *p* = 0.0311) or clear margin groups (12.1%; *p* = 0.0039). Similarly, the percentage of patients with positive margins who presented nodal disease was also significantly greater (60%) than in the close margin (42.8%; *p* = 0.0135) or clear margin groups (26.5%; *p* = 0.0001). The positive margin group also had a significantly higher percentage of patients with ENE (29.3%) versus the close margin (17.2%; *p* = 0.0334) and clear margin groups (7.2%; *p* = 0.001). Other risk factors, including smoking status, higher grade of tumour, PNI, and LVI, were similar among these three subgroups (Table 3).

Figure 3, Figure 4 and Figure 5 show the Kaplan–Meier survival curves (DFS) for patients with local (Figure 3), regional (Figure 4), and distant (Figure 5) disease according to margin status (positive, close, clear). DFS was significantly worse in the positive margin group. No significant survival differences were observed between the close and clear margin groups.

Table 4 shows the risk factors for local, regional, and distant recurrence according to margin status. In the positive margin group, the risk of local recurrence was highest in patients with LVI (HR = 3.33), advanced disease (HR = 3.01), ENE (HR = 2.53), and nodal disease (HR = 2.45). Those same four risk factors, together with PNI (HR = 3.02) and smoking (HR = 2.8), were also significant risk factors for regional recurrence. The only significant risk factor for distant recurrence was smoking status (HR = 3.93) (Table 4).

In patents with close margins, the risk of local recurrence was highest in patients with nodal disease (HR = 3.42), advanced disease (HR = 2.83), and LVI (HR = 2.71). For regional recurrence, those same three factors plus PNI (HR = 4.66) were significant. For distant recurrence, nodal disease, advanced disease, and ENE (HR = 3.66) were all significant (Table 4).

Finally, in patients with clear margins, the only variables that were significant for a higher risk of local recurrence were nodal disease (HR = 4.08) and advanced disease (HR = 3.94). None of these risk factors were significant for regional or distant recurrence (Table 4).

### 3.7. Cut-Off Point to Differentiate between Survival Outcomes in Patients with Close Margins

In the close margin group, we performed an analysis of the time-dependent receiver operating characteristic (ROC) curve based on the Youden index in an attempt to identify a cut-off point that would indicate better survival. This analysis was performed separately for the deep and mucosal margins (Table 5).

The optimal deep margin cut-off size for DFS (area under the curve [AUC] = 0.58) and OS (AUC = 0.63) was a margin > 3 mm. No significant cut-off point was found for the mucosal margin. Figure 6 and Figure 7 show the DFS and OS Kaplan–Meier curves grouped according to margin status (positive margins, deep margin ≤ 3 mm, deep margin > 3 mm, and clear margins). For DFS, the difference between positive margins and close margins > 3 mm was statistically significant (33.8% vs. 80.2% respectively, *p* < 0.0001; HR = 3.31). Similarly, there were significant differences in the 5-year DFS between the groups with positive and clear margins (33.8% vs. 64.2%, *p* = 0.0002; HR = 2.17) and between those with a deep margin ≤ 3 vs. >3 mm (52.6% vs. 80.2%, *p* = 0.006; HR = 2.1). In terms of OS at 5 years, significant differences were observed between the following: positive vs. ≤3 mm margins (31.8% vs. 41.7% respectively, *p* = 0.004; HR = 1.7); positive vs. >3 mm margins (31.8% vs. 71.3% respectively, *p* < 0.0001; HR = 3.94); positive vs. clear margins (31.8% vs. 65.4%, *p* = 0.0004; HR = 2.15); and ≤ vs. >3 mm (41.7% vs. 71.3%, *p* = 0.0003; HR = 2.32) (Table 5).

Table 6 shows the risk of local, regional, and distant recurrence according to margin status (positive, close, clear) and deep margin distance (≤3 vs. >3 mm). As the table shows, there were significant differences in local recurrence between the following: positive vs. >3 mm margins (HR = 4.74); positive vs. clear margins (HR = 2.74); and ≤3 vs. >3 mm margins (HR = 2.45). For regional recurrence, significant differences were observed between the following: positive vs. >3 mm margins (HR = 4.9); positive vs. clear margins (HR = 2.55); and ≤3 vs. >3 mm margins (HR = 2.55). For distant recurrence, significant differences were observed for the following: positive vs. >3 mm margins (HR = 7.19); positive vs. clear margins (HR = 7.51); ≤3 vs. >3 mm margins (HR = 2.72); and ≤3 mm vs. clear (HR = 2.84). No survival differences were observed between patients with >3 mm margins and clear margins.

## 4. Discussion

In the close margin subgroup, survival outcomes were significantly better (AUC = 0.63) in patients with a deep margin > 3 mm vs. those with smaller margins (≤3 mm) (Table 5). At 5 years, the DFS and OS (80.2% and 71.3%) in this subgroup with deep margins > 3 mm were similar to those observed in the clear margin group (64.2% and 65.4%, respectively). By contrast, patients with a smaller deep margin (≤3 mm) had worse 5-year DFS and OS rates (52.6% and 41.7%), which were comparable to the positive margin group (33.8% and 31.8%, respectively).

In our study, the variable that had the greatest impact on the risk of distant failure was positive surgical margins, a finding that contrasts with the results reported in other published studies, most of which have found that nodal involvement is the main risk factor [22,23,24]. Crucially, none of the clinical or tumour-related variables evaluated in this study had any significant impact on regional or distant disease in patients with clear surgical margins. Similarly, none of these factors were significant for distant recurrence in the positive margin group. These findings suggest that other factors (immune deficiency, genetic factors, and adverse histologic risk factors), which are not routinely studied, likely play a role in treatment failure.

In oral cancer, margin status is a well-known risk factor for recurrence. However, the effect size of positive margins versus other strong risk factors (e.g., nodal status and advanced local disease) is controversial [25]. In addition, although close margins are considered to be a risk factor, the true risk has not been well described. In fact, as we have shown in this study, the risk appears to depend in large part on the size of the margin. Consequently, this lack of clarity complicates decision-making with regards to adjuvant treatment in patients with close margins. In the study by Mitchell et al. [26], the mean survival in patients with involved margins was only half of that observed in patients with clear margins (11.4 vs. 25.4 years, respectively). In that study, patients with close margins also had significantly worse survival (15.6 years) than those with clear margins. However, this difference between patients with close and clear margins was not confirmed in our study (5-year OS 54% vs. 65,4%, *p* = 0.669; DFS 63.8% vs. 64.2%, *p* = 0.8399, respectively). Although 5-year OS and DFS were significantly worse in the positive margin group (31.8% and 33.8%, respectively; *p* < 0.0001), we did not observe any significant differences between the close and clear margin groups. Binahmed et al. reported similar findings, with no significant differences between the close and clear margin groups [27]. However, those authors defined a “close” margin as ≤2 mm from the inked resection margin. A systematic review carried out by Young et al. concluded that patients with margins <1 mm had a three-fold greater risk of local recurrence compared to those with clear margins [28]. However, that study did not compare the hazard ratio for margin status to other known risk factors.

A study in Taiwan involving more than 15,000 patients included in the Taiwan Cancer Registry database found that three factors—advanced nodal status (HR = 5.30), positive surgical margins (HR = 2.10), and advanced T stage (HR = 2.08)—were associated with the greatest risk of recurrence in a multivariate analysis [29]. However, it is difficult to compare the findings of that study to ours because they used cancer-specific survival (CSS), whereas we used DFS and OS and we also grouped our results according to local, regional, and distant recurrence. Notwithstanding these differences, our multivariate analysis showed that nodal disease (HR = 3.39), advanced disease (HR = 3.37), and positive surgical margins (HR = 2.02) were all associated with a significantly increased risk of local recurrence. The most significant risk factors for regional failure were nodal status and advanced disease, followed by PNI, LVI and ENE.

Another finding of this study was that positive surgical margins (HR = 5.6) had the greatest impact on the risk of distant failure, followed by nodal disease (HR = 4.01). This finding is particularly interesting because nodal involvement is the main risk factor in most other studies. For example, Zubair et al. evaluated 187 patients with recurrent disease, 46 of whom developed distant recurrence. The main risk factors were nodal disease (HR = 23.04 for pN2c) and depth of invasion (HR = 14.03 for >10 mm) [30]. Zanoni et al. found that the neutrophile-to-lymphocyte (NLR) ratio—which we did not evaluate in our study—and nodal disease (>5 metastatic nodes) were the strongest risk factors (HR = 3.1 and HR = 2.06, respectively) for distant recurrence [31].

In the positive margin group, the variables with the greatest risk of local and regional recurrence were, respectively, LVI (HR = 3.33) and ENE (HR = 4.45). None of the other variables were significantly associated with the risk of distant recurrence. In patients with close surgical margins, nodal disease had the greatest impact on local and distant recurrence (HR = 3.42 and HR = 9.16, respectively). The presence of PNI (HR = 4.66) was associated with the greatest risk of regional recurrence. In the clear margin subgroup, nodal disease was associated with a significant increase in risk of local recurrence (HR = 4.08).

Interestingly, none of the risk factors evaluated in this study were significantly associated with either regional or distant recurrence. This finding is important because it suggests that recurrence must be attributable to risk factors (e.g., NLR, pattern of invasion, tumour budding, etc.) that were not evaluated in our study. Zanoni et al. found that NLR was the greatest risk factor for the development of distant recurrence, indicating that an impaired immune system is a key risk factor that should be considered when planning adjuvant treatment and surveillance [31]. Kligerman et al. examined 161 patients with oral cancer but excluded patients with certain risk factors (PNI, LVI, and positive surgical margins). That study evaluated certain adverse pathologic risk factors including the pattern of invasion, tumour budding, and tumour infiltrating lymphocytes. In the multivariate analysis, tumour budding, higher T stage, and neck dissection were all significantly associated with recurrence [17].

Although numerous previous studies [10,12,13,14,31,32] have also subdivided patients with close margins in an effort to determine the impact of the margin size on outcomes, the results to date have been heterogenous. Moreover, none of those studies differentiated between mucosal and deep margins. For example, the study by Singh et al. used two margin cut-off points, i.e., >2 mm and >7 mm, to determine their impact on the mean locoregional-free survival [32]. Lin et al. [29] took 5 mm margins as a reference point, finding that margins < 4 mm were associated with significantly worse CSS. However, that study did not differentiate between mucosal and deep margins. Zanoni et al. used ROC curves to identify a cut-off point to differentiate between close margins, finding that survival outcomes in patients with a margin > 2.2 mm were similar to those observed in patients with clear margins (AUC = 0.671). Nonetheless, that study only measured locoregional-free survival at 2 years [14]. Fowler et al. evaluated more than 400 patients with oral cancer, finding that patients with margins > 1 mm had similar outcomes to those with clear margins (>5 mm) [15].

### Strengths and Limitations

This study has several limitations. First, the retrospective design is a limitation, with all the limitations inherent to this type of study. Another limitation, related to the retrospective design, is that the decision to administer adjuvant radiotherapy was individualized after presentation to a multidisciplinary tumour board, which could have introduced bias and thus influenced the results. By contrast, the main strength of this study is that it was a single-institution study with a highly standardized treatment regimen with reliable follow-up. Importantly, to our knowledge, this is the first study to analyze outcomes according to both deep and mucosal margins.

## 5. Conclusions

This study has two important findings. First, among the patients with close margins, survival was significantly better (and comparable to the clear margin subgroup) in those with a deep margin > 3 mm. This suggests that adjuvant radiotherapy or chemoradiotherapy may not always be necessary in patients with close margins (but only in well-selected cases). Another relevant finding is that positive surgical margins—not nodal involvement—had the greatest impact on the risk of distant failure. Given the potential clinical and treatment implications of these findings, more research—ideally through prospective studies—is warranted.

## Figures and Tables

**Figure 1 cancers-16-01488-f001:**
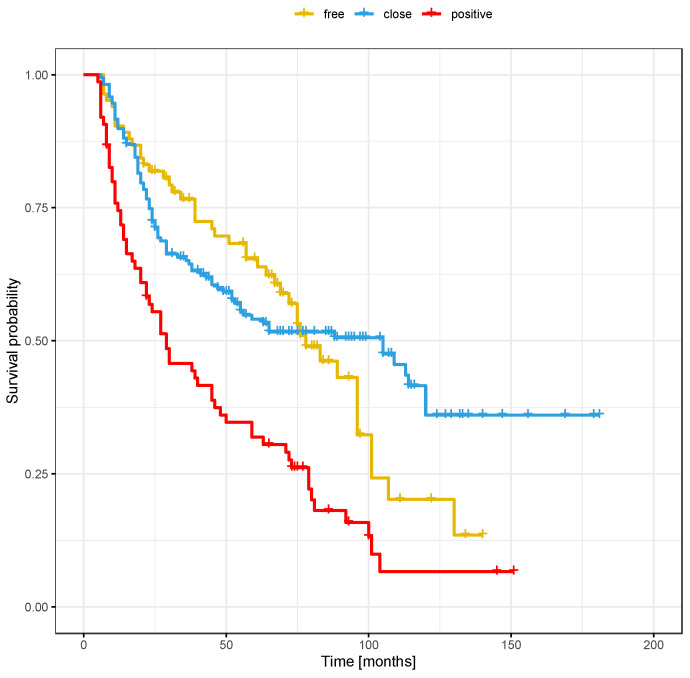
Overall survival stratified by margin status.

**Figure 2 cancers-16-01488-f002:**
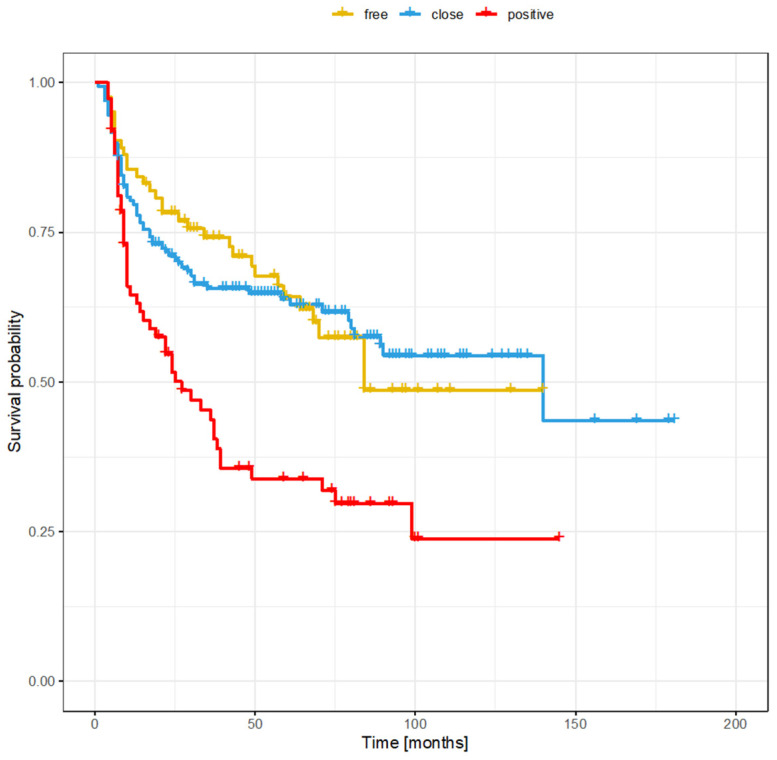
Disease-free survival stratified by margin status.

**Figure 3 cancers-16-01488-f003:**
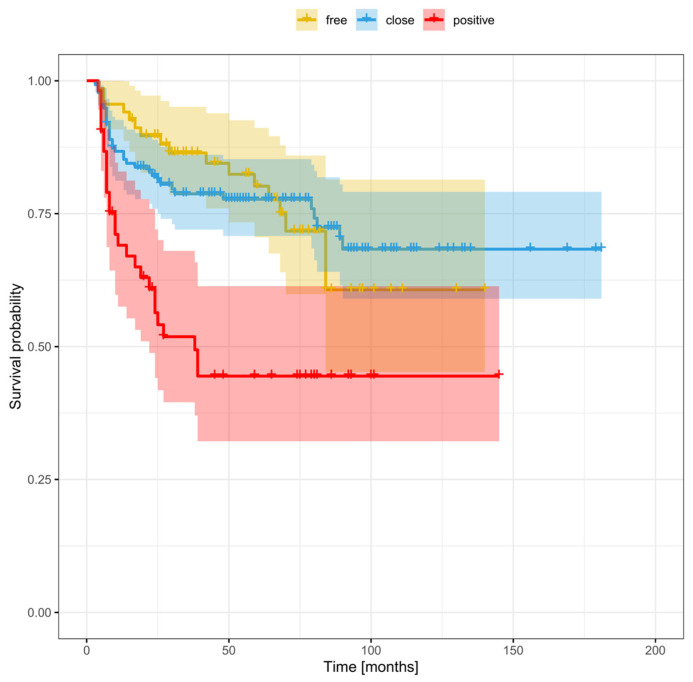
Disease-free survival stratified by margin status in patients with local recurrence.

**Figure 4 cancers-16-01488-f004:**
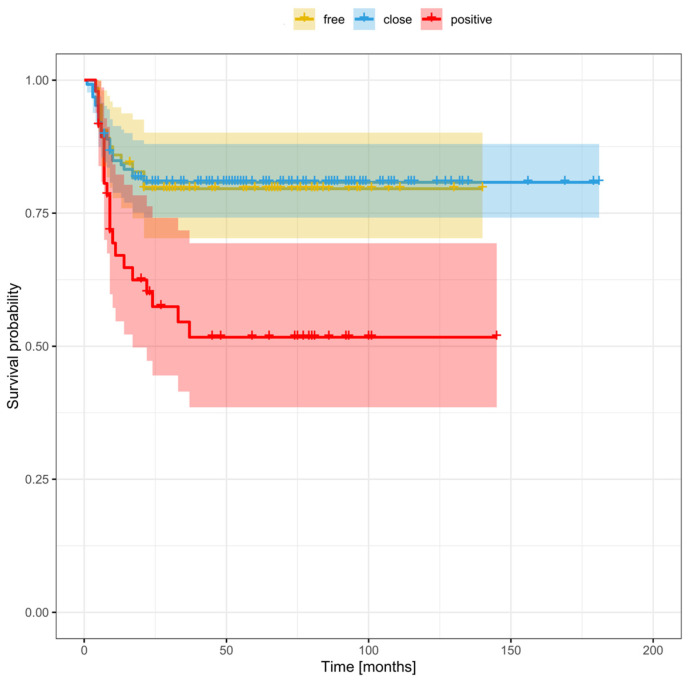
Disease-free survival stratified by margin status in patients with regional recurrence.

**Figure 5 cancers-16-01488-f005:**
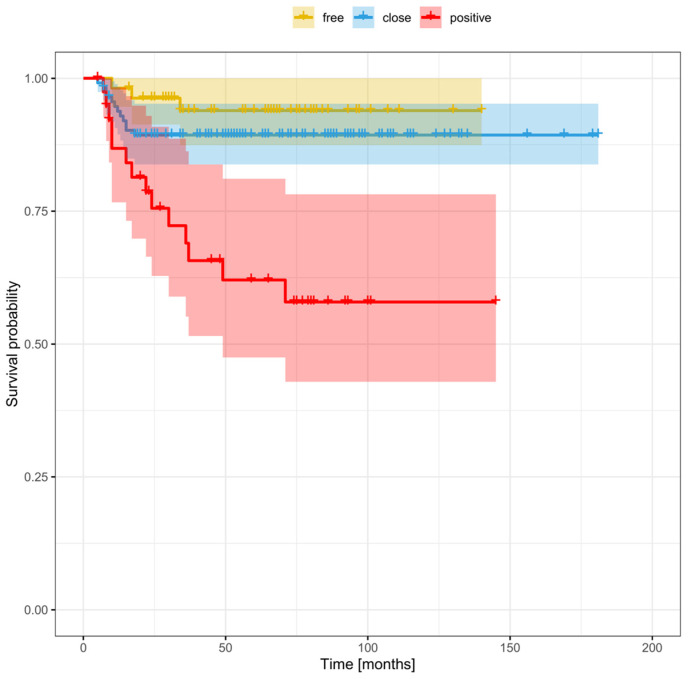
Disease-free survival stratified by margin status in patients with distant recurrence.

**Figure 6 cancers-16-01488-f006:**
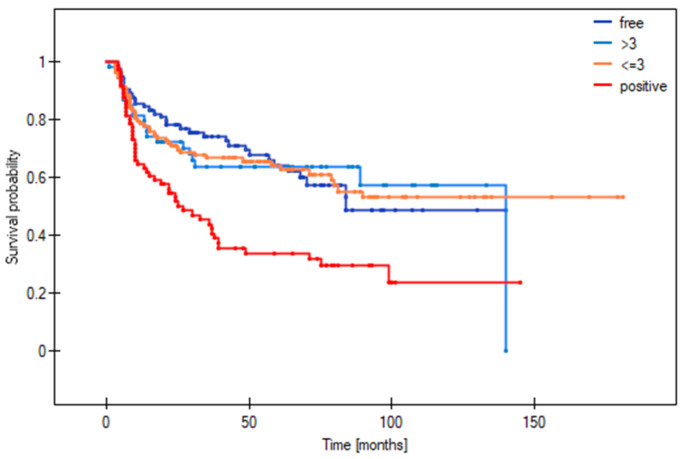
Disease-free survival stratified by margin status, including a smaller (≤3 mm) and larger close margin (>3 mm).

**Figure 7 cancers-16-01488-f007:**
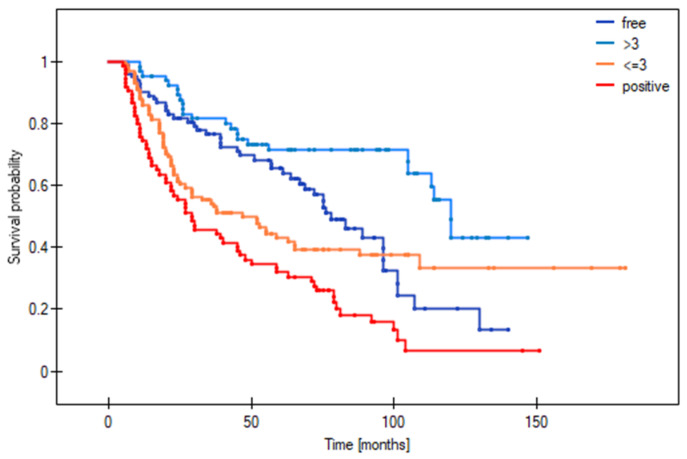
Overall survival stratified by margin status, including a smaller (≤3 mm) and larger close margin (>3 mm).

**Table 1 cancers-16-01488-t001:** Clinical and demographic features of the study population.

Variable	Patients (n = 326)	%
Sex		
Male	210	64.4
Female	116	35.6
Mean age, years (range)	59.1 (23–97)	
Tobacco use		
Never	129	39.5
Active	197	60.5
Alcohol use		
Never/occasionally (less than 20 g of 100% alcohol per week)	281	86.1
Alcohol abuse	45	13.9
Location		
Tongue	155	47.5
Floor of mouth	111	34.0
Buccal mucosa	29	8.9
Retromolar trigone	8	2.5
Mandibular gingiva	21	6.5
Maxillary gingiva	2	0.6
Grade		
1	67	20.5
2	208	63.8
3	51	15.7
Clinical T stage		
Early (T1/T2)	258	79.1
Advanced (T3/T4)	68	20.9
Clinical N stage		
N0	187	57.3
N+	139	42.7
Tumour stage		
Early (I/II)	156	47.8
Advanced (III/IV)	170	52.2
PNI	40	12.2
LVI	27	8.2
ENE	57	17.4
Margins		
Positive (<1 mm)	83	25.5
Close (1–5 mm)	168	51.5
Clear (>5 mm)	75	23%
Adjuvant treatment		
Radiotherapy	263	80.6
Chemoradiotherapy	85	26
Recurrence	147	45.1
Local	53	16.2
Regional	33	10.1
Locoregional	17	5.2
Distant	21	6.4
Locoregional + distant	8	2.4
Second primary tumour	15	4.6

Abbreviations: PNI, perineural invasion; LVI, lymphovascular invasion; ENE, extranodal extension.

**Table 2 cancers-16-01488-t002:** Univariate and multivariate analysis of risk factors for recurrent disease.

Risk Factor	Univariate Analysis		Multivariate Analysis	
	HR (95% CI)	*p* Value	HR (95% CI)	*p* Value
**Local recurrence**				
Grade (3 vs. 1 + 2)	1.71 (0.97–3.01)	0.0671	0.9 (0.5–1.63)	0.7265
T stage (advanced vs. early)	1.75 (1.07–2.87)	**0.0260**	1.3 (0.77–2.2)	0.3343
Nodal disease (N+ vs. N0)	3.54 (2.23–5.64)	**<0.0001**	3.39 (1.9–6.03)	**<0.0001**
Stage (advanced vs. early)	3.32 (2.02–5.44)	**<0.0001**	3.37 (1.84–6.16)	**0.0001**
PNI	2.09 (1.15–3.79)	**0.0156**	1.5 (0.79–2.85)	0.2207
LVI	2.94 (1.55–5.59)	**0.0001**	1.9 (0.93–3.88)	0.0774
ENE	1.91 (1.11–3.27)	**0.0189**	1.41 (0.78–2.53)	0.2514
Adjuvant treatment	0.64 (0.34–1.21)	0.1654	1.78 (0.8–4.0)	0.1605
Margins				
positive	2.75 (1.5–5.06)	**0.0011**	2.02 (1.02–4.01)	**0.0446**
close	1.02 (0.57–1.83)	0.9464	0.96 (0.51–1.81)	0.9071
clear	ref	ref	ref	ref
Smoking status (active vs. never)	0.58 (0.37–0.91)	**0.017**	0.67 (0.41–1.08)	0.0997
Alcohol abuse	1.25 (0.68–2.32)	0.4704	1.14 (0.57–2.27)	0.7065
**Regional recurrence**				
Grade (3 vs. 1 + 2)	2.16 (1.21–3.85)	**0.0088**	1.11 (0.59–2.09)	0.7439
T stage (advanced vs. early)	1.5 (0.83–2.07)	0.1776	1.05 (0.59–2.09)	0.8948
Nodal disease (N+ vs. N0)	3.66 (2.14–6.26)	**<0.0001**	3.85 (1.84–8.04)	**0.0003**
Stage (advanced vs. early)	3.6 (2.02–6.42)	**<0.0001**	3.93 (1.89–8.18)	**0.0003**
PNI	3.58 (2.03–6.31)	**<0.0001**	2.74 (1.47–5.13)	**0.0016**
LVI	3.67 (1.94–6.94)	**0.0001**	1.99 (0.99–3.99)	**0.0052**
ENE	2.82 (1.63–4.89)	**0.0002**	2.1 (1.11–3.98)	**0.0228**
Adjuvant treatment	0.87 (0.45–1.68)	0.678	2.9 (0.89–4.28)	0.0876
Margins				
positive	2.56 (1.28–5.12)	**0.0078**	1.91 (0.85–4.32)	0.118
close	0.96 (0.49–1.88)	0.898	0.89 (0.42–1.88)	0.7593
clear	ref		ref	
Smoking status (active vs. never)	0.53 (0.32–0.88)	**0.0151**	0.54 (0.32–0.93)	**0.025**
Alcohol abuse	1.02 (0.48–2.16)	0.9531	0.66 (0.27–1.61)	0.363
**Distant recurrence**				
Grade (3 vs. 1 + 2)	1.19 (0.41–3.43)	0.7451	0.62 (0.21–1.82)	0.381
T stage (advanced vs. early)	0.95 (0.36–2.49)	0.9178	0.52 (0.19–1.39)	0.1902
Nodal disease (N+ vs. N0)	5.94 (2.62–13.44)	**<0.0001**	4.01 (1.63–9.89)	**0.0025**
Stage (advanced vs. early)	3.79 (1.68–8.56)	**0.0014**	2.26 (0.94–5.42)	0.0687
PNI	1.31 (0.4–4.34)	0.6537	0.63 (0.16–2.4)	0.4933
LVI	4.74 (1.93–11.66)	**0.0007**	5.65 (1.78–17.95)	**0.0033**
ENE	3.46 (1.61–7.46)	**0.0015**	2.19 (0.92–5.23)	**0.076**
Adjuvant treatment	0.13 (0.02–0.97)	**0.0471**	0.52 (0.06–4.64)	0.5561
Margins				
Positive	7.53 (2.16–26.22)	**0.0015**	5.6 (1.41–22.19)	**0.0142**
Close	1.99 (0.56–7.04)	0.2874	1.05 (0.27–4.05)	0.9491
Clear	ref		Ref	
Smoking status (active vs. never)	1.1 (0.5–2.43)	0.8046	1.15 (0.51–2.62)	0.7374
Alcohol abuse	1.75 (0.71–4.31)	0.2218	1.09 (0.41–2.9)	0.8699

Abbreviations: HR, hazard ratio; CI, confidence interval; PNI, perineural invasion; LVI, lymphovascular invasion; ENE, extranodal extension. Bold value stands for *p* value that is statistically significant.

**Table 3 cancers-16-01488-t003:** Distribution of clinical and tumour-related factors by margin status (positive, close, clear).

Clinical or Tumour-Related Factors	Positive Margins (n = 83)	Close Margins (n = 168)	Clear Margins (n = 75)	Positive vs. Close	Positive vs. Clear	Clear vs. Close
	n (%)			*p* Value		
Active smoker	42 (56)	106 (63.1)	49 (59.1)	0.6998	0.6998	0.6998
Advanced T stage	25 (33.3)	33 (19.6)	10 (12.1)	**0.0311**	**0.0039**	0.133
Nodal disease	45 (60)	72 (42.9)	22 (26.5)	**0.0135**	**0.0001**	**0.0135**
Advanced disease	50 (67.6)	92 (54.8)	27 (32.5)	0.0623	**0.0014**	**<0.0001**
PNI	13 (17.3)	20 (11.9)	7 (8.4)	0.3808	0.1438	0.4037
LVI	8 (10.7)	16 (9.5)	3 (3.6)	0.7827	0.1438	0.1438
ENE	22 (29.3)	29 (17.3)	6 (7.3)	**0.0334**	**0.001**	**0.0334**
Adjuvant treatment	74 (98.7)	141 (83.9)	48 (57.8)	**0.0009**	**<0.0001**	**<0.0001**

Abbreviations: PNI, perineural invasion; LVI, lymphovascular invasion; ENE, extranodal extension. Bold value stands for *p* value that is statistically significant.

**Table 4 cancers-16-01488-t004:** Risk factors for local, regional, and distant recurrence according to margin status.

** *Positive Surgical Margins* **						
	**Local Recurrence**		**Regional Recurrence**		**Distant Recurrence**	
	**HR (95% CI)**	***p* Value**	**HR (95% CI)**	***p* Value**	**HR (95% CI)**	***p* Value**
**Advanced T stage**	1.67 (0.79–3.68)	0.1758	1.8 (0.76–4.28)	0.1844	0.76 (0.21–2.72)	0.4326
**Nodal disease**	2.45 (1.07–5.64)	**0.0346**	3.54 (1.28–9.78)	**0.0147**	2.53 (0.84–7.61)	0.0992
**Advanced disease**	3.01 (1.14–7.97)	**0.0268**	3.16 (1.06–9.43)	**0.0395**	1.57 (0.53–4.71)	0.4174
**PNI**	2.36 (0.88–6.32)	0.0868	3.02 (1.15–7.89)	**0.0245**	0.99 (0.13–7.64)	0.9943
**LVI**	3.33 (1.1–10.07)	**0.0335**	4.05 (1.43–1.48)	**0.0084**	-	-
**ENE**	2.53 (1.12–5.72)	0.0261	4.41 (1.82–10.71)	**0.001**	3.06 (0.93–10.04)	0.0648
**Adjuvant treatment**	-	0.9337	-	0.9795	-	0.9825
**Smoking status**	1.44 (0.67–3.09)	0.3487	2.8 (1.02–7.69)	**0.0466**	3.93 (1.09–14.23)	**0.0341**
** *Close Surgical Margins* **						
	**Local Recurrence**		**Regional Recurrence**		**Distant Recurrence**	
	**HR (95% CI)**	***p* Value**	**HR (95% CI)**	***p* Value**	**HR (95% CI)**	***p* Value**
**Advanced T stage**	1.67 (0.78–3.58)	0.1879	1.19 (0.45–3.2)	0.7239	0.92 (0.2–4.21)	0.9912
**Nodal disease**	3.42 (1.69–6.94)	**0.0006**	3.7 (1.58–8.66)	**0.0025**	9.16 (2.01–41.82)	**0.0043**
**Advanced disease**	2.83 (1.35–5.92)	**0.0059**	5.75 (1.96–16.83)	**0.0014**	5.74 (1.26–26.21)	**0.024**
**PNI**	2.11 (0.87–5.1)	0.0994	4.66 (2.03–10.7)	**0.0003**	1.06 (0.14–8.18)	0.9585
**LVI**	2.71 (1.12–6.58)	**0.0272**	3.64 (1.44–9.19)	**0.0063**	-	-
**ENE**	1.44 (0.63–3.31)	0.3912	2.0 (0.83–4.82)	0.1235	3.66 (1.16–11.53)	**0.027**
**Adjuvant treatment**	1.63 (0.71–3.75)	0.2482	2.1 (0.83–5.3)	0.1156	-	0.9795
**Smoking status**	0.4 (0.2–0.81)	0.01	0.24 (0.1–0.56)	**0.001**	0.81 (0.24–2.68)	**0.7268**
** *Clear Surgical Margins* **						
	**Local Recurrence**		**Regional Recurrence**		**Distant Recurrence**	
	**HR (95% CI)**	***p* Value**	**HR (95% CI)**	***p* Value**	**HR (95% CI)**	***p* Value**
**Advanced T stage**	1.01 (0.23–4.42)	0.9933	0.57 (0.07–4.36)	0.586	-	0.9843
**Nodal disease**	4.08 (1.57–10.61)	**0.004**	2.95 (0.96–9.04)	0.0587	9.96 (0.9–110.25)	0.061
**Advanced disease**	3.94 (1.48–10.48)	**0.0059**	1.93 (0.63–5.91)	0.2493	5.97 (0.54–65.85)	0.1447
**PNI**	1.4 (0.32–6.2)	0.6551	2.19 (0.49–9.9)	0.307	5.56 (0.5–61.62)	0.1621
**LVI**	5.87 (0.73–47.41)	0.0965	3.3 (0.43–25.56)	0.2524)	-	-
**ENE**	0.82 (0.11–6.24)	0.8501	2.35 (0.52–10.62)	0.2675	5.43 (0.49–60.24)	0.1678
**Adjuvant treatment**	0.37 (0.12–1.12)	0.0792	0.69 (0.22–2.1)	0.5077	0.5 (0.05–5.33)	0.5717
**Smoking status**	0.66 (0.25–1.71)	0.3942	0.29 (0.09–0.95)	**0.0415**	0.3 (0.03–3.31)	0.326

Abbreviations: HR, hazard ratio; CI, confidence interval; PNI, perineural invasion; LVI, lymphovascular invasion; ENE, extranodal extension. Bold value stands for *p* value that is statistically significant.

**Table 5 cancers-16-01488-t005:** Disease-free survival and overall survival stratified according to the deep margin distance (≤ vs. >3 mm).

Margin Status	5-Year DFS	5-Year OS
Positive	33.8%	31.8%
Deep margin ≤3 mm	52.6%	41.7%
Deep margin >3 mm	80.2%	71.3%
Clear	64.2%	65.4%
	Deep margin ≤ 3 mm	
*p* value *	**<0.0001**	**<0.0001**
	Hazard ratio (95% CI)	
>3 mm vs. clear	0.66 (0.42–1.03)	0.54 (0.37–0.8)
≤3 mm vs. clear	1.38 (0.9–2.13)	1.26 (0.87–1.84)
positive vs. clear	**2.17 (1.34–3.52)**	**2.15 (1.39–3.31)**
≤3 mm vs. >3 mm	**2.1 (1.35–3.28)**	**2.32 (1.59–3.39)**
positive vs. >3 mm	**3.31 (2.02–5.41)**	**3.94 (2.55–6.1)**
positive vs. ≤3 mm	1.57 (0.98–2.53)	**1.7 (1.11–2.61)**

Abbreviations: DFS, disease-free survival; OS, overall survival. * Log-rank test. Bold value stands for *p* value that is statistically significant.

**Table 6 cancers-16-01488-t006:** Local, regional, and distant recurrence risk stratified by margin status.

	Local Recurrence Risk	Regional Recurrence Risk	Distant Recurrence Risk
	Deep margin ≤ 3		
	**Hazard ratio (95% CI)**		
*p*-value *	**<0.0001**	**0.0007**	**<0.0001**
>3 mm vs. clear	0.58 (0.32–1.05)	0.52 (0.26–1.06)	1.04 (0.39–2.82)
≤3 mm vs. clear	1.41 (0.79–2.54)	1.33 (0.67–2.62)	**2.84 (1.08–7.48)**
Positive vs. clear	**2.74 (1.37–5.45)**	**2.55 (1.17–5.57)**	**7.51 (2.45–23.04)**
≤3 vs. >3 mm	**2.45 (1.35–4.46)**	**2.55 (1.26–5.14)**	**2.72 (1.02–7.24)**
Positive vs. >3 mm	**4.74 (2.36–9.55)**	**4.9 (2.2–10.89)**	**7.19 (2.32–22.27)**
Positive vs. ≤3 mm	1.93 (0.98–3.83)	1.92 (0.89–4.17)	2.64 (0.87–8.02)

* Log-rank test. Bold value stands for *p* value that is statistically significant.

## Data Availability

For the data supporting the reported results, please contact the corresponding author.

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
