# Peer review of "A Matter of Margins in Oral Cancer—How Close Is Enough?"

_cancers, 2024, doi:10.3390/cancers16081488_

Round 1

Reviewer 1 Report

Comments and Suggestions for Authors

The purpose of the present study was to determine the risk factors for recurrence by margin status and to identify a margin cut-off point for improved survival in patients with close margins.

A retrospective review of adult patients treated surgically for primary oral cavity squamous cell cancer were registered from 2009 to 2021.

To main objective of the present study was to determine the risk factors for recurrence by margin status and to identify a margin cut-off point for improved survival in patients with close margins.

Please provide bibliography for this two paragraphs-Patients with close margins represent a highly heterogenous subgroup with some patients 61 presenting more adverse clinicopathological features than others. In fact, the data show 62 that some patients with close margins achieve clinical outcomes that are similar to those 63 observed in patients with clear margins, whereas others experience outcomes that are 64 more in line with those found in patients with involved margins. While numerous factors 65 play a role in determining treatment outcomes in patients with close margins, there is little 66 doubt that the surgical margin is a key factor. However, as the findings of the studies 67 described above indicate, margins < 5 mm may be sufficient in some cases. In this regard, 68 it would be valuable to identify a cut-off point that would differentiate between patients 69 with close margins in terms of better or worse survival. 70 In patients with oral cancer, the risk factors for local, regional, and distant recurrence 71 according to margin status have not been well-established. In this context, we conducted 72 the present retrospective study to determine the risk factors for recurrence according to 73 margin status and to identify a margin cut-off point for improved survival in patients with 74 close margins.

Why the study was until 2021?

Please rephrase this part-Interestingly, the variable that had the greatest impact on the risk of distant failure 271 was positive surgical margins, a finding that contrasts with the results reported in other 272 published studies, most of which have found that nodal involvement is the main risk fac- 273 tor17–19.

Terms as -Interestingly, An unexpected finding, are not appropriate for a scientific articles. Please rephrase.

Comments on the Quality of English Language

moderate

Author Response

Dear Reviewer,

first of all, I would like to thank you for all your comments and time you took to evaluate our manuscript.

Moving into details, bibliography for the mentioned paragraphs has been provided.   

The study was until 2021 due to precise follow up of all patients was done until the end of 2023. We have mentioned it in the Material and Methods section that only patients with a minimum follow up of 24 months were included in the study. 

Suggested words/ sentences have been rephrased, thank you for this comment

Kind regards

Reviewer 2 Report

Comments and Suggestions for Authors

The article “A Matter of Margins in Oral Cancer – How Close Is Enough?” it is very interesting, and I have some considerations to make with the intention of improving the manuscript.

Material and methods:

-        Clearly indicate the inclusion criteria

-        You should include the reference to the 7th edition of the American Joint Committee on Cancer (AJCC)

Results:

-        Why did you not include the alcohol variable, indicated in Table 1, in the univariate and multivariate analysis?

On the other hand, what do you consider to be an “occasional” drinker?

-        Was the variable smoking status and "close surgical margins" and "clear surgical margins" analyzed? You have not been showed in table 4

Conclusion

-        Maybe this conclusion is a little risky. Line 364 “This suggests that adjuvant radiotherapy or chemoradiotherapy may not always be necessary in patients with close margins

Thank you

Author Response

Dear Reviewer,

first of all I would like to express my gratitude for for work and time spent while evaluating our manuscript. I'm sure that your comments will highly improve our article.

In methodology section the inclusion criteria have been revised. A reference to 7th edition of AJCC has also been added.

The effect of alcohol on the risk of local, regional and distant recurrence has been added to results section (Table 2). Additionally occasional alcohol consumption in our study was defined as less than 20 grams of 100% alcohol per week (this sentence has been added to Table 1. )

In results section the effect of smoking status on the risk of recurrence based on margin status was added to Table 4. 

The sentence in conclusion section has been rephrased (added "only in well-selected cases")

Kind regards

Reviewer 3 Report

Comments and Suggestions for Authors

The authors performed a study in order to to determine the risk factors for recurrence by margin status and to identify a margin cut-off point for improved survival in patients with close margins. The article is of interested and well written although minor changes are needed:

- Please reduce the number of tables

-  Any information about histological type?

- In the skin it is known that some types of cancers (such as basal cell carcinoma) can show an aggressive behavior according some clinical and histologic predictive factors (as reported in this recent article Clinical and Dermoscopic Factors for the Identification of Aggressive Histologic Subtypes of Basal Cell Carcinoma. Front Oncol. 2021 ), can you use some aspects from this study about oral mucosa and adapt them for your study? This can improve your evidences.

- Can you summarize the main systemic treatment performed by your patients?

Author Response

Dear Reviewer,

first of all I would like to express my gratitude for for work and time spent while evaluating our manuscript. I'm sure that your comments will highly improve our article.

It's really difficult to meet your suggestion in terms of the number of tables. The reason for that is all the information provided in each table is being presented in results and later discussed in discussion section.

The only histological type analyzed in the study is oral mucosal squamous cell carcinoma. The information is presented in Meterial and Methods Section. 

I have read the article that you have suggested and it's really interesting but it's really hard to cite it in a manuscript that focusses only on mucosal squamous cell cancer. 

The systemic treatment regimen was added to Material and Methods Section.

Kind regards!